# On the role of domain aspect ratio in the westward intensification of wind-driven surface ocean circulation

Kaushal Gianchandani[1], Hezi Gildor[1], and Nathan Paldor[1]

[1]Fredy & Nadine Herrmann Institute of Earth Sciences, Hebrew University of Jerusalem, Edmond J. Safra Campus - Givat Ram, Jerusalem, 9190401, Israel

**Correspondence:** Nathan Paldor (nathan.paldor@mail.huji.ac.il)

**Abstract.** The two seminal studies on westward intensification, carried out by Stommel and Munk over 70 years ago, are revisited to elucidate the role of the domain aspect ratio (i.e. meridional to zonal extents of the basin) in determining the transport of the western boundary current (WBC). We examine the general mathematical properties of the two models by transforming them to differential problems that contain only two parameters — the domain aspect ratio and the non-dimensional damping (viscous) coefficient. Explicit analytical expressions are obtained from solutions of the non-dimensional vorticity equations and verified by long-time numerical simulations of the corresponding time-dependent equations. The analytical expressions as well as the simulations, imply that in Stommel's model both the domain aspect ratio and the damping parameter contribute to the non-dimensional transport of the WBC. However, the transport increases as a cubic power in the aspect ratio and decreases linearly with the damping coefficient. On the other hand, in Munk's model the WBC's transport increases linearly with the domain aspect ratio, while the damping coefficient plays a minor role only. This finding is employed to explain the weak WBC in the South Pacific. The decrease in transport of the WBC for small domain aspect ratio results from the decrease in Sverdrup transport in the basin's interior because the meridional shear of the zonal velocity cannot be neglected as an additional vorticity term.

## 1  Introduction

As was noted by Henry Stommel, in the opening sentence of his seminal 1948 study "Perhaps the most striking feature of the general oceanic wind-driven circulation is the intense crowding of streamlines near the western borders of the oceans." These strong and narrow poleward directed currents, often referred to as "western boundary currents" (WBCs), counterbalance the weak and wide equatorward (Sverdrup) flow in the interior of the basin. In the North Atlantic this current is the Gulf Stream, and it was known to oceanographers and explorers for a few centuries — see Stommel (1958) for a historical review. Similar WBCs exist in other basins as well and these include the Kuroshio in the North Pacific and the Brazil current in the South Atlantic. These currents transport large amount of heat from low to high latitudes, thus playing an important role in the climate

system. The winds overlying, though, are easterlies along the equator (the Trade winds) and westerlies around 40° N. There are no strong northward winds along the western boundaries of the ocean basins and, as is well understood now, the WBCs are

not obviously correlated with the overlying wind patterns. Interestingly, two such WBCs lie in the Pacific viz. the Kuroshio and the East Australian Current (EAC). Both the Kuroshio and the EAC are centered close to 26° latitude in their respective hemispheres, are driven by similar wind stresses, and are adjacent to a $\sim$ 2000 km long coastline. Despite these structural similarities, the maximal volumetric transport of the Kuroshio current is 55 Sv (1 Sv = $10^6$ m$^3$s$^{-1}$) (Qiu, 2019) whereas that of the EAC is around 30 Sv (Archer et al., 2017). The maximum velocity that EAC attains is also substantially smaller than

that of the Kuroshio (Campisi-Pinto et al., 2020).

Stommel, apparently in his first oceanography paper (Stommel, 1948, hereafter referred to as S48) was the first to formulate a simple, yet comprehensive, mathematical model of the WBCs [see e.g. Kunzig (1999)]. S48 is now regarded as a seminal paper in theoretical physical oceanography (e.g. http://empslocal.ex.ac.uk/people/staff/gv219/classics.d/oceanic.html). S48's model probably provides the simplest explanation for the existence of WBCs: in this linear and frictional model on the $\beta-$plane

the ocean is taken to be a flat bottom rectangle forced by a cos(latitude)-dependent zonal wind pattern. Walter Munk further extended this work to a different frictional (viscous) parameterization and a more general form of the wind stress (Munk, 1950, hereafter referred to as M50).

In the last 70 years, both models have been modified and extended to further explore the phenomenon of westward intensi-fication in different settings or to evaluate the importance of different specific processes and terms in the governing equations

(Munk and Carrier, 1950; Veronis, 1966a, b; Pedlosky, 2013; Vallis, 2017, and references therein).

As in S48 and M50, a large number of these subsequent studies employed the dimensional form of the governing equations which are the time-independent rotating linearized shallow water equations compounded by friction and forcing. These di-mensional models include numerous parameters: the zonal and meridional extents of the basin; either the coefficient of linear drag (i.e. the coefficient in the Rayleigh frictional term) or the kinematic eddy viscosity (i.e. the coefficient in parameterization

of the viscous term); the amplitude (and possibly meridional structure) of the wind stress; the gradient of Coriolis frequency ($\beta-$effect). On the other hand, a few studies (Welander, 1976; Bye and Veronis, 1979) employed the alternate, concise, ap-proach of non-dimensionalising the governing equation (or the vorticity equation) to investigate the depth averaged wind-driven ocean circulation. The non-dimensional approach not only simplifies the problem by reducing the number of dimensional pa-rameters in the model to fewer non-dimensional ones but also brings out some salient features associated with the problem

which are difficult to unveil in the dimensional formulation.

By employing a non-dimensional approach, Welander (1976) successfully identified a zonally uniform regime in both S48's and M50's models of wind-driven ocean circulation and using the same approach, Bye and Veronis (1979) derived a correc-tion to the Sverdrup transport in S48's model. The aforementioned studies highlighted the importance of the ratio between meridional and zonal extents of the basin as one of the two fundamental parameters in both S48's and M50's models. The

aim of this study is to further elaborate on the role of the domain aspect ratio (defined here as the ratio between the basin's meridional and zonal extents) in S48's and M50's models of westward intensification. In particular, we examine the role of domain aspect ratio in the transport of the WBC as was first hypothesized by Bye and Veronis (1979) in the context of S48's

model, "... the tendency of north-south diffusive processes to be more significant in basins with a large (*small in the present scaling*) aspect ratio makes sense physically and may play a quantitative role in the transport of the western boundary current."
We also examine the relevance of our results to the observed difference in strengths of the five WBCs in the world ocean.

The paper is organized as follows. Section 2 outlines our proposed scaling [which is slightly different from the one employed in Welander (1976); Bye and Veronis (1979)] that reduces the number of parameters in the vorticity equations corresponding to S48's and M50's models from five dimensional ones to two non-dimensional ones — one of which is the domain aspect ratio (the other is damping). The solution for the stream function in the two cases is outlined and using this we obtain the expression
for the non-dimensional transport of the WBC in both S48's and M50's models. The applicability of the analytical expression of transport for relevant values of the model parameters is validated in Section 3 by simulating the time-dependent equations. We discuss the results and conclude in Section 4. We also note that there were some typos in the expressions of zonal velocity and sea surface height (but not the stream function itself) in S48 and for completeness, we list them in Appendix A. These typos do not change the scientific conclusions drawn in S48.

## 2 The two-parameter differential problems, their solutions and the transports of the WBC

### 2.1 S48's non-dimensional counterpart

S48's dimensional vorticity equation for the spatial structure of the stream function, $\psi$, is given by:

$$r\nabla^2\psi + \beta\frac{\partial\psi}{\partial x} = \tau_0\frac{\pi}{\rho_0 H_0 L_y}\sin\left(\frac{\pi y}{L_y}\right) \tag{1}$$

where $r$ is the Rayleigh friction coefficient, $\beta$ is the meridional gradient of Coriolis frequency and $\tau_0$ is the amplitude of wind-
stress. The operator $\nabla^2$ is the two dimensional Laplacian, $H_0$ is the mean depth of the barotropic ocean with density $\rho_0$, $L_y$ is the meridional dimension (and $L_x$ is the zonal dimension) of the basin. The velocity components in the zonal and meridional directions, $u$ and $v$, are related to the stream function via: $u = \frac{\partial\psi}{\partial y}$ and $v = -\frac{\partial\psi}{\partial x}$.

We begin by scaling (1) as follows: $x$ (the zonal coordinate) on $L_x$; $y$ (the meridional coordinate) on $L_y$ and $\psi$ on $\gamma\beta L_y^3$ where $\gamma = \tau_0\frac{\pi}{L_y}\left(\frac{L_x}{\rho_0 H_0 \beta^2 L_y^3}\right)$ is the non-dimensional amplitude of the wind stress curl. With this scaling the non-dimensional form
of S48's vorticity equation is:

$$\frac{\epsilon}{\delta^2}\nabla^2\psi + \frac{\partial\psi}{\partial x} = \sin(\pi y) \tag{2}$$

where

$$\epsilon = \frac{r}{\beta L_x}, \quad \nabla^2 = \delta^2\frac{\partial^2}{\partial x^2} + \frac{\partial^2}{\partial y^2}. \tag{3}$$

From this point onwards, both the variables and the operators in the differential equation(s) are non-dimensional while
dimensional quantities will be accompanied by an asterisk (*). Here $\nabla^2$ is the non-dimensional Laplacian, $\delta = \frac{L_y}{L_x}$ is the ratio of meridional and zonal extents of the basin (refereed to as the domain aspect ratio) and $\epsilon$ is the non-dimensional width of the WBC (and also a proxy of the damping). The definition of the non-dimensional stream function implies that the zonal

velocity and the meridional velocity are given by, $u = \dfrac{\partial \psi}{\partial y}$ and $v = -\delta \dfrac{\partial \psi}{\partial x}$, respectively. It is evident from (2) and (3) that the two parameters, $\epsilon$ and $\delta$, govern the structure of the flow in the basin. The no normal flow conditions at the basin's boundaries

mandate that the stream function $\psi$ satisfies the boundary conditions: $\psi(x,0) = \psi(1,y) = \psi(x,1) = \psi(0,y) = 0$. We note that the term domain aspect ratio in the present study, $\delta = \dfrac{L_y}{L_x}$, is the inverse of the domain aspect ratio used in Bye and Veronis (1979) who derived a non-dimensional equation similar to (2).

As has been stated earlier, the non-dimensional formulation lumps the five dimensional parameters in S48's model — zonal and meridional extent of the basin, gradient of Coriolis frequency, wind stress amplitude and Rayleigh friction coefficient —

into just two non-dimensional ones: $\epsilon$ and $\delta$ [both of which appear only in the first term of (2)]. Following S48, an explicit expression of the solution for $\psi$ in (2) is given by:

$$\psi(x,y) = \frac{\delta^2}{\epsilon \pi^2} \sin(\pi y)(pe^{Ax} + qe^{Bx} - 1) \tag{4}$$

where

$$p = \frac{1 - e^B}{e^A - e^B}$$

$q = 1 - p$

and

$$A = -\frac{1}{2\epsilon} + \frac{\pi}{\delta}\sqrt{1 + \frac{\delta^2}{4\pi^2 \epsilon^2}},$$

$$B = -\frac{1}{2\epsilon} - \frac{\pi}{\delta}\sqrt{1 + \frac{\delta^2}{4\pi^2 \epsilon^2}}.$$

As is evident from (4), the spatial structure of the stream function is controlled by both $\epsilon$ and $\delta$. Panels (a) and (c) of Fig.

1 depict the stream functions for two $\epsilon$-regimes of S48's model: (i) weak damping [$\epsilon \leq \delta^2$] and (ii) strong damping [$\epsilon > \delta^2$]. For $\epsilon \leq \delta^2$, the solution $\psi$ given by (4) becomes linear in $x$ and thus can satisfy only one boundary condition out of two. This solution is commonly assumed to approximate the exact solution for $\psi$ in the frictionless interior of the basin while a different approximation applies in the narrow, frictional, boundary layer adjacent to $x = 0$. Fig. 1(a) depicts this narrow boundary layer for $\epsilon = 0.1\delta^2$ where the stream function first decreases fast with $x$ for small $x$ and then increases slowly with $x$ for large $x$.

For $\epsilon > \delta^2$, the solution, $\psi$, is symmetric about $x = \dfrac{1}{2}$ and can satisfy the two boundary conditions, $\psi(0,y) = 0 = \psi(1,y)$. This is demonstrated in the symmetric stream function depicted in Fig. 1(c) for $\epsilon = 10\delta^2$. The explicit expressions of $\psi$ in the two ranges of $\epsilon$ are given in Appendix B.

In S48's model, we define the transport of the WBC as the product of its width, $\epsilon$, and the average of the meridional velocity,

$v = -\delta \dfrac{\partial \psi}{\partial x}$, between the western edge of the basin, $x = 0$, and $x = \epsilon$ evaluated along $y = \dfrac{1}{2}$ i.e. $Tr = \epsilon \left( \dfrac{1}{\epsilon} \displaystyle\int_0^\epsilon -\delta \dfrac{\partial \psi}{\partial x}\Big|_{y=\frac{1}{2}} dx \right)$.

The integral in the definition of the transport, $Tr$, simplifies to the product of the domain aspect ratio and the difference in the values of the stream function evaluated at $x = 0$ and $x = \epsilon$ along $y = \dfrac{1}{2}$, i.e. $Tr = \delta \left[ \psi\left(0, \dfrac{1}{2}\right) - \psi\left(\epsilon, \dfrac{1}{2}\right) \right]$. Substituting the

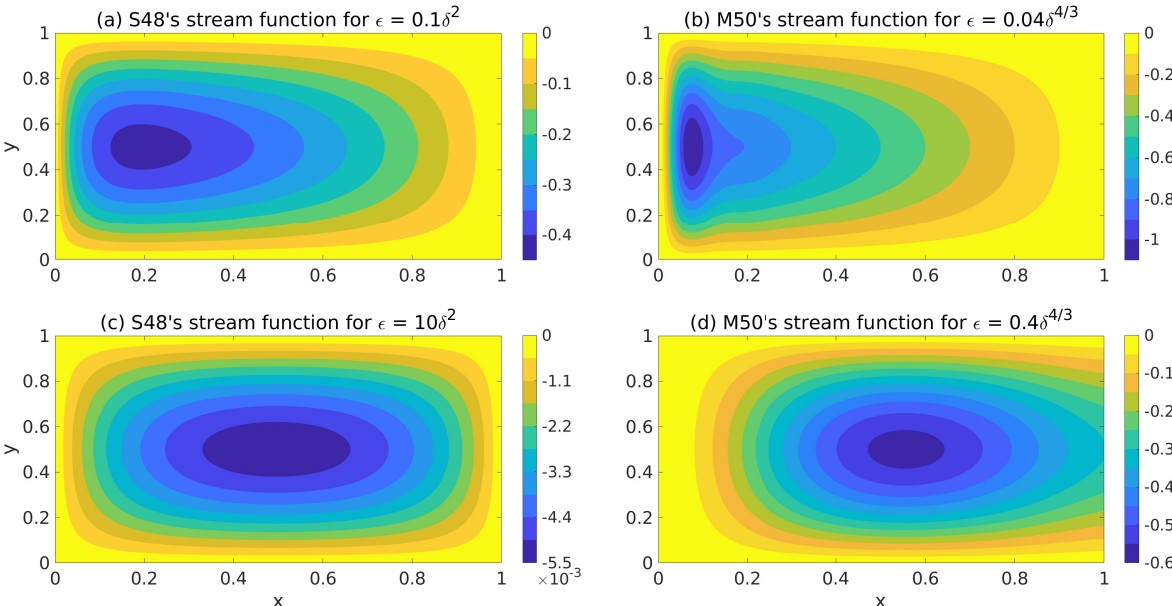

**Figure 1.** The stream functions in different $\epsilon$ regimes of S48's and M50's models for $\delta = 2\pi/10$: (a) and (b) weak damping [$\epsilon \leq \delta^2$ in S48's model and $\epsilon \leq 0.1\delta^{4/3}$ in M50's model] — there exists a narrow fast flowing current along the western edge of the basin. (c) and (d) strong damping [$\epsilon > \delta^2$ in S48's model and $\epsilon > 0.1\delta^{4/3}$ in M50's model] — the stream function is (nearly) symmetric about $x = 0.5$ which indicates that there is no westward intensification.

boundary condition $\psi\left(0, \dfrac{1}{2}\right) = 0$ and using the explicit solution (4) yields:

$$Tr = \frac{\delta^3}{\epsilon\pi^2}(1 - pe^{A\epsilon} - qe^{B\epsilon}). \tag{5}$$

This expression will be compared below to its counterpart in M50's model and will be compared in section 3 with transports
calculated by numerical simulations.

Here, we note that the definition of the WBC's width is somewhat arbitrary and for definiteness we choose it to be $\epsilon$ [as in Welander (1976); Bye and Veronis (1979); Vallis (2017)]. However, the conclusions drawn in this study are independent of the precise definition; for instance, the width of the WBC can also be defined as the value of $x$ at which the stream function reaches an extremum. According to this definition the WBC's width, $\epsilon'$ equals $\sim 5\epsilon$ and the corresponding transport is given by
$Tr' = \dfrac{\delta^3}{5\epsilon\pi^2}(1 - pe^{5A\epsilon} - qe^{5B\epsilon})$. Both, expressions of $Tr$ and $Tr'$, yield that the transport of the WBC in S48's model varies as $\sim \dfrac{\delta^3}{\epsilon}$.

## 2.2 M50's non-dimensional counterpart

The non-dimensional counterpart of M50's vorticity equation, obtained by employing the scaling proposed in this study in a similar manner to that of S48 [refer to Munk (1950) for the dimensional equation], is given by:

$$-\frac{\epsilon^3}{\delta^4}\nabla^4\psi + \frac{\partial\psi}{\partial x} = \sin(\pi y) \tag{6}$$

where

$$\epsilon = \frac{1}{L_x}\left(\frac{\mu}{\beta}\right)^{1/3}, \quad \nabla^4 = \delta^4\frac{\partial^4}{\partial x^4} + 2\delta^2\frac{\partial^4}{\partial x^2 \partial y^2} + \frac{\partial^4}{\partial y^4} \tag{7}$$

where $\mu$ is the (dimensional) horizontal eddy viscosity coefficient. We note that contrary to (2), the sign in front of the first term in (6) is negative. This dissimilarity arises because, unlike the parametrization in S48, in M50's model the damping is parametrized by the two dimensional bilaplacian operator. Also, in addition to stream function vanishing at the edges of the basin another set of boundary condition has to be specified to solve the $4^{th}$ order equation (6). The additional boundary conditions employed by M50 originate from the inclusion of lateral viscosity which implies that there should be no tangential flow at the basin's edges i.e. $\frac{\partial\psi}{\partial x}\Big|_{x=0,1} = \frac{\partial\psi}{\partial y}\Big|_{y=0,1} = 0$. Following the mathematical steps in M50 yields the following approximate solution of (6):

$$\psi = -\sin(\pi y)\left[1 - x + \epsilon e^{(x-1)/\epsilon} - e^{-(x/2\epsilon)}\xi(\epsilon)\right] \tag{8}$$

and $\xi(\epsilon) = \left[\cos\left(\frac{\sqrt{3}x}{2\epsilon}\right) + \frac{1-2\epsilon}{\sqrt{3}}\sin\left(\frac{\sqrt{3}x}{2\epsilon}\right)\right]$.

Panels (b) and (d) of Fig. 1 depict the stream function for small and large damping in M50's model. For large damping the stream function shown in Fig. 1(d) is not entirely symmetric about $x = \frac{1}{2}$. Also, unlike the behavior of the stream function in S48's model, the stream function in M50's model skews more towards the eastern boundary with the increase in damping. This, less than optimal, behavior of the stream function in M50's model occurs because the stream function does not vanish identically along the eastern boundary and is, instead, a function of $\epsilon$ itself (although, for small $\epsilon$, the zonal velocity there is small compared to the rest of the basin).

We turn now to the estimation of the WBC's transport in M50's model. As was done in S48's model, this transport is also defined as the product of the boundary layer width ($\epsilon$) and the mean meridional velocity of the current between $x = 0$ and $x = \epsilon$ along $y = \frac{1}{2}$. Following the arguments laid out in the previous section [see the paragraph above (5)] an expression for transport can be obtained by multiplying the domain aspect ratio by the difference of the stream function values between $x = 0$ and $x = \epsilon$ along $y = \frac{1}{2}$. Furthermore, substituting the boundary condition $\psi\left(0, \frac{1}{2}\right) = 0$ yields $Tr = -\delta\psi\left(\epsilon, \frac{1}{2}\right)$. Evaluating $\psi$ in (8) at $\left(\epsilon, \frac{1}{2}\right)$ for $\epsilon \ll 1$ yields the following simplified expression for the WBC's transport in M50's model:

$$Tr = \delta\left(1 - e^{(-1/2)}\left[\cos\left(\frac{\sqrt{3}}{2}\right) + \frac{1-2\epsilon}{\sqrt{3}}\sin\left(\frac{\sqrt{3}}{2}\right)\right]\right). \tag{9}$$

As anticipated by Bye and Veronis (1979), the transport of the WBC in S48's model [given by (5)] is governed by both
damping ($\epsilon$) and domain aspect ratio ($\delta$). However, in M50's model the dependence of the WBC's transport on the two pa-
rameters is strikingly different: the transport is governed primarily by $\delta$ and is weakly dependent on $\epsilon$. In the next section we
validate these claims using (dimensional) numerical simulations and then apply our results to the present-day world ocean.

## 3    Numerical simulations and application to the world ocean

The numerical simulations described below were carried out using the time-dependent, forced-dissipative, rotating shallow
water equation (SWE) dimensional solver that was successfully used in previous studies. The solver employs the finite dif-
ference method to solve SWEs on the $\beta-$plane and the simulations are carried out on an Arakawa C grid with leapfrog time
difference scheme. Though the solver can include nonlinear terms, these terms were neglected in the present application. The
reader should refer to Gildor et al. (2016) and Shamir et al. (2019) for a more detailed description of the solver.

The simulations presented here were carried out in a barotropic ocean with the same characteristics as in S48 i.e. on an
equatorial $\beta-$plane ($f_0 = 0$), forced by a wind stress that varies as $-\tau_0 \cos\left(\dfrac{\pi y^*}{L_y}\right)$. Three of the dimensional parameters
remained fixed in all the simulations presented below — the gradient of Coriolis frequency (given by $\beta = 2 \times 10^{-11}$ m$^{-1}$s$^{-1}$),
the zonal extent of the basin ($L_x$ = 10000 km) and the amplitude of the prescribed forcing ($\tau_0 = 0.2$ Nm$^{-2}$). The other two
dimensional parameters in the two WBC models i.e. the damping coefficients [Rayleigh friction coefficient ($r$) in S48's model
and horizontal eddy viscosity ($\mu$) in M50's model] and the meridional extent of the basin ($L_y$) are varied to examine the effect
of $\epsilon$ and $\delta$ on the transport. We note that keeping $\tau_0$ fixed and varying $L_y$ will yield different values of $\gamma$ in the simulation,
however, since we scale our $\psi^*$ on $\gamma \beta L_y^3$ and only look at the non-dimensional transport we do not have to account for the
effects of changes in $\gamma$. The results are consistent with what one would obtain by keeping only $\beta$ and $L_x$ fixed and varying
$\tau_0$ along with the damping coefficients and $L_y$ to keep $\gamma$ constant. The boundary conditions are: the (dimensional) zonal and
meridional velocities vanish along the basin's meridional and zonal boundaries respectively, i.e. $u^*|_{y^*=0,L_y} = v^*|_{x^*=0,L_x} = 0$.
The numerical solver is integrated until a steady state is reached. The steady state of the time-dependent simulations is defined
as the state at which the dependent variables in the SWEs [dimensional zonal velocity ($u^*$), meridional velocity ($v^*$) and sea
surface height ($\eta^*$)] cease to evolve for sufficiently long time.

Panels (a) and (c) in Fig. 2 depict the numerically obtained, non-dimensional stream function $\left(\psi = \dfrac{\psi^*}{\gamma \beta L_y^3}\right)$ in the steady
state for the dimensional parameters as in S48's model (and the corresponding value of $\delta$ is noted above these panels) while
panels (b) and (d) in Fig. 2 depict the numerically obtained, non-dimensional $\psi$ in the steady state for the parameters relevant
to M50's model (and here too the corresponding value of $\delta$ is noted above these panels). The WBC's width is kept constant
in all four panels and is noted in the caption. The reader should note that the meridional extent ($L_y$) of the basin shown in
panels (a) and (b) of Fig. 2 is $2\pi \times 1000$ km, whereas, the meridional extent of the basin shown in panels (c) and (d) in Fig. 2
is $L_y = \pi/4 \times 1000$ km. In all four cases the shape of the stream function is very similar to the steady non-dimensional stream
functions shown in Fig. 1 [panels (a) and (b)]. We note that for the given values of ($\epsilon$, $\delta$), the $\psi$s obtained from dividing the

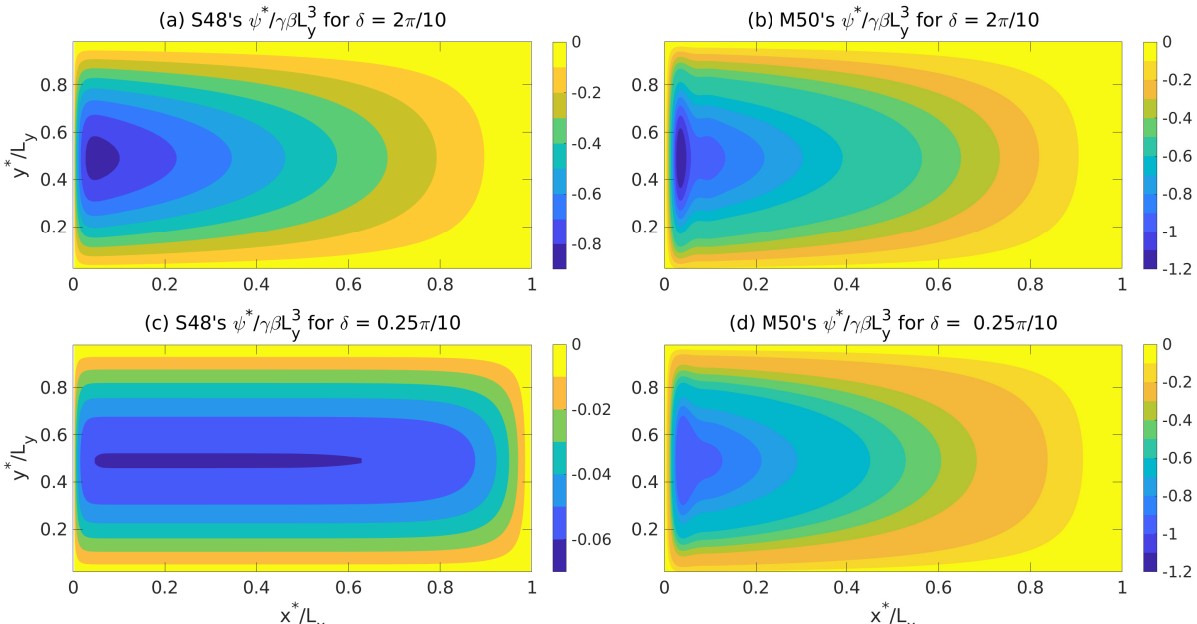

**Figure 2.** Numerically obtained, non-dimensional stream functions for $\epsilon = 0.01$ and $L_x = 10000$ km. (a) S48's model with $\delta = 2\pi/10$, (b) M50's model with $\delta = 2\pi/10$. Panels (c) and (d) are the same as (a) and (b) but for $\delta = 0.25\pi/10$ i.e. the meridional extent of the basin is one-eighth of that in (a) and (b). Note the different colorbars in panels (a) and (c).

numerically calculated $\psi^*$s by the corresponding values of $\gamma\beta L_y^3$ are in agreement with the $\psi$ calculated analytically for the same values of $(\epsilon, \delta)$ using (4) for S48's model and (8) for M50's model.

Fig. 2 depicts that in both S48's and M50's models, for a fixed value of $\epsilon = 0.01$ (damping and the width of the WBC), the gradient of the stream function increases with $\delta$. The higher zonal gradient of the stream function near the western boundary yields a larger meridional velocity, thus increasing the transport of the WBC (given by the product of width and average meridional velocity of the WBC). Clearly, $\delta$ exercises control over the transport of the WBC and hence cannot be ignored.

Fig. 3 compares the analytic and numerically computed values of the non-dimensional transport $(Tr)$ of the WBC in S48's and M50's models as a function of $\epsilon$ for several values of $\delta$. The solid lines denote the analytic value of $Tr$ obtained from the expressions given by (5) and (9). The 'numerical transport' of the WBC is obtained by taking the product of $\delta$ and $-\dfrac{\psi^*(\epsilon, \frac{1}{2})}{\gamma\beta L_y^3}$. Here, $\psi^*\left(\epsilon, \dfrac{1}{2}\right)$ is the value of the steady state dimensional stream function at $\left(\epsilon, \dfrac{1}{2}\right)$ obtained from the results of the numerical simulation for a given set of parameters which correspond to a certain $(\epsilon, \delta)$.

As is evident by Fig. 3(a), the analytic and numerically calculated non-dimensional transports of the WBC in S48's model are in good agreement. Fig. 3(b) shows that the analytic transports of the WBC in M50's model are nearly independent of $\epsilon$ and are governed primarily by $\delta$. The numerically calculated transports of the WBC depicted by the dashed lines in Fig. 3(b) show a similar dependence on $\delta$ but in contrast to the approximate analytic expression these transports vary slightly with $\epsilon$. We also

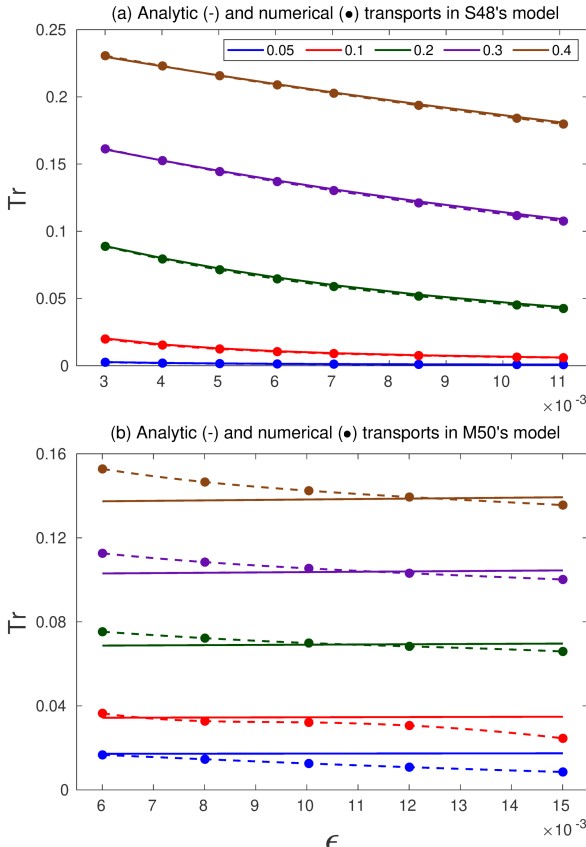

**Figure 3.** Comparison between analytically (solid lines) and numerically (dots) calculated values of transport ($Tr$) as a function of $\epsilon$ for different values of $\delta$ in (a) S48's and (b) M50's models. The dashed lines depict the cubic spline interpolated curves between the numerically calculated transports (dots).

note that there is a discernible difference between the analytically estimated and numerically calculated values of transport in M50's model for nearly all values of $(\epsilon, \delta)$. This is because the expression for the stream function for M50's model, [i.e. (8)] only crudely approximates the actual stream function.

Fig. 4 depicts the non-dimensional transport of the WBC in S48's [panel (a)] and M50's models [panel (b)] as contours on $(\epsilon, \delta)$ plane. The contours were obtained by interpolating (using the cubic spline method) between the numerically calculated values of the WBC's transport as shown in Fig. 3 (plus a few additional values at low $\epsilon$ in S48's model). As is evident from Fig. 4(a), the non-dimensional transport is a function of both $\epsilon$ and $\delta$ in S48's model. On the other hand, Fig. 4(b) shows that the transport of the WBC is only weakly dependent on $\epsilon$ and is governed primarily by $\delta$ (the contours are nearly parallel to the abscissa). The position of the different WBCs in the $(\epsilon, \delta)$ parameter space is marked with different symbols and the errorbars account for the inaccuracies in the assigned values of the zonal and meridional extents of the basins. The details of how the

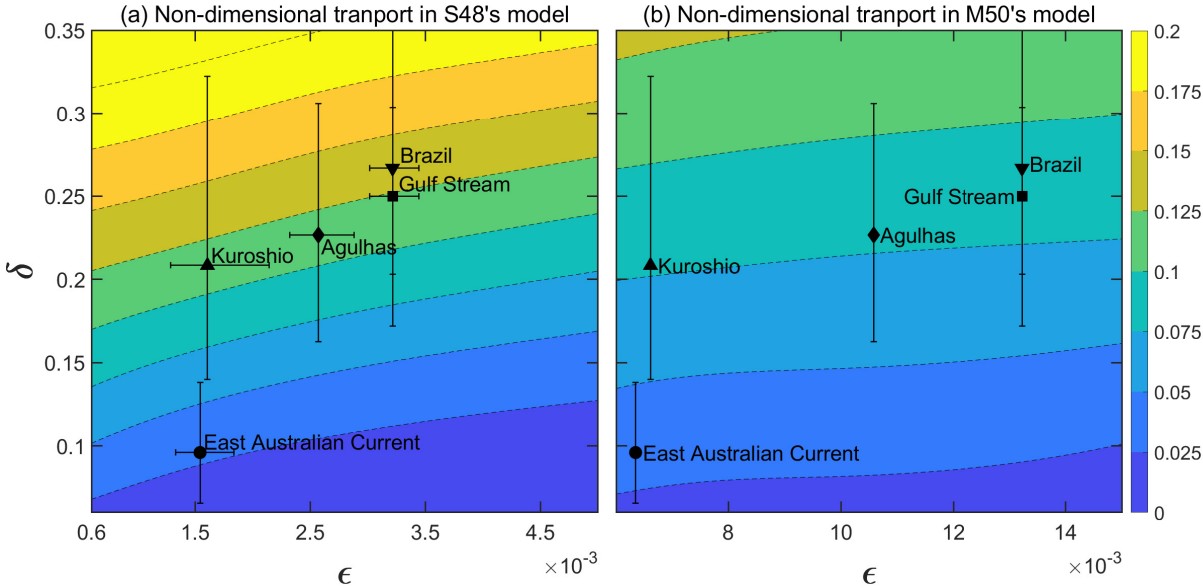

**Figure 4.** The non-dimensional transport of the western boundary current (WBC) as a function of $\epsilon$ and $\delta$ in (a) S48's model and (b) M50's model. Contours depicted are obtained from numerical simulations for fixed $L_x = 10000$ km, $\beta = 2 \times 10^{-11}$ m$^{-1}$s$^{-1}$ and by varying $r$ between 1/100 days$^{-1}$ and 1/10 days$^{-1}$ in S48's model [panel (a)] and $\mu = 10000$ m$^2$s$^{-1}$ in M50's model [panel (b)]. The different WBCs in the world ocean are depicted with different symbols and the error bars denote the possible variability of parameters that can occur because of an error in estimating the zonal and meridional extents of the basins that contain the WBC. The width of each of the WBCs, $\epsilon$, is estimated using $r = 1/30$ days$^{-1}$ [panel (a)] and $\mu = 10000$ m$^2$s$^{-1}$ [panel (b)] and $\beta = 2 \times 10^{-11}$ m$^{-1}$s$^{-1}$. The zonal extent, $L_x$, of the basin containing the particular WBC is given in Table C1. The error in $\epsilon$ is not accounted for in (b) because the WBC's transport in M50's model is (nearly) independent of $\epsilon$. The East Australian Current's (EAC) non-dimensional transport, as calculated from both S48's and M50's models, is less than the other four WBCs. The uncertainty in $\delta$ for the Brazil Current extends up to 0.375 in both models, the contours have been restricted to better resolve the boundary currents. The range between which the non-dimensional transport varies is similar in both the models.

irregular basins in the world ocean are approximated with rectangles are discussed in Appendix C. The error in $\epsilon$ has been omitted from 4(b) because the WBC's transport in M50's model is nearly independent of $\epsilon$. Despite the large uncertainty in the damping parameters (relevant in S48's model only) and domain aspect ratios (relevant in both S48's and M50's models) of the five WBCs, the non-dimensional transport of the East Australian Current is distinctly smaller than that of the other WBCs.

## 4 Summary and Discussions

Since the introduction of the S48's and M50's models about 70 years ago, numerous theoretical and numerical investigations have been carried out to further explore the characteristics of westward intensification (Munk and Carrier, 1950; Stommel, 1958; Hogg and Johns, 1995; Pedlosky, 2013; Vallis, 2017, and references therein). Both S48's and M50's dimensional models

clearly bring out the contribution of each source of vorticity: damping, planetary gradient and wind forcing in producing the characteristic east-west asymmetry of the flow in a basin. However, it is difficult to quantify the contribution of each of the five dimensional parameters ($L_x$, $L_y$, $\beta$, $\tau_0$ and $r/\mu$) to the transport of WBC, using the dimensional models. A better alternative is to combine several dimensional parameters to yield a system with fewer non-dimensional parameters as was employed by, for example, Welander (1976) to identify a zonally uniform regime in ocean circulation and by Bye and Veronis (1979) to identify

the correction to the Sverdrup transport in context of S48's original model.

    In this article, we address the issue raised by Bye and Veronis (1979) regarding the effect of domain aspect ratio on the WBC's transport by providing explicit expressions of the non-dimensional transport in both S48's and M50's models. These expressions are then benchmarked against numerical simulations of the time dependent, forced-dissipative, rotating shallow water equations. Both the analytic expressions and steady state simulations show that the WBCs' transports depend on both

$\epsilon$ and $\delta$ in S48's model, however, a change in $\delta$ has a stronger effect on $Tr$ when compared to a change in $\epsilon$ $\left( Tr \sim \dfrac{\delta^3}{\epsilon} \right)$. In contrast, the transport of the WBC in M50's model is nearly independent of $\epsilon$ and is governed primarily by $\delta$ ($Tr \sim \delta$).

    In the traditional description of the S48 model the flow is decomposed into two parts: A slow, anti-cyclonic flow in the inner-basin where the velocities are tiny so frictional effects can be neglected and a return boundary flow where the frictional vorticity associated with the zonal shear of the poleward directed velocity, balances the planetary vorticity advected by this

velocity. According to this paradigm the WBC simply returns the frictionless equatorward Sverdrup transport of the inner-basin so its transport is independent of the friction coefficient and since the (dissipation) Laplacian term does not affect the Sverdrup interior flow, the transport of the WBC should also be independent of the domain aspect ratio. The present study demonstrates that the assumption of small damping, $\epsilon \ll 1$, implies that only the term $\epsilon \dfrac{\partial^2 \psi}{\partial x^2}$ of the Laplacian in (2) can be neglected in this limit while the second term, $\dfrac{\epsilon}{\delta^2} \dfrac{\partial^2 \psi}{\partial y^2}$, cannot be neglected in the interior solution when $\delta^2 \sim O(\epsilon)$. The implication of our

analysis is that the Sverdrup interior flow depends on $\delta$ for sufficiently small $\delta$ and therefore so does the (return) transport of the WBC.

    To appreciate this subtle issue one should compare a square basin, where $\delta = 1$, with a narrow and long "channel-like" basin where $\delta \ll 1$. In a square basin, the classical approach of equating $\dfrac{\partial \psi}{\partial x}$ to $\sin(\pi y)$ in the inner basin works well since the north-south gradient of the zonal velocity (represented by $\dfrac{\partial^2 \psi}{\partial y^2}$) is small and can be neglected from the interior solution. However,

in a "channel-like" ocean this quantity is large and cannot be neglected from the balance of terms in the interior solution. An examination of the three vorticity terms in the interior [$\dfrac{\partial \psi}{\partial x}$, $\dfrac{\partial^2 \psi}{\partial y^2}$ and $-\sin(\pi y)$] clarifies that the transport of the WBC (as well as the equatorward transport in the interior) in a "channel-like" ocean should be smaller compared to a square ocean since the meridional shear of the zonal velocity lowers the vorticity induced by the curl of the wind stress. In an extreme "channel-like" ocean with $\dfrac{\epsilon}{\delta^2} \gg 1$ the only term that can balance $\dfrac{\partial^2 \psi}{\partial y^2}$ is $\dfrac{\partial \psi}{\partial x}$ that implies a strong, equatorward directed velocity. Indeed, as

was shown by Welander (1976) for a small domain aspect ratio, a boundary layer develops along the basin's eastern boundary in which the strong current flows equatorward.

    In M50's model the vorticity balance of the interior is more involved since the bilaplacian dissipation operator ($\nabla^4$) has 3 terms, each of which with a coefficient of different power of $\delta$. Thus, the distinction between terms associated with the

inner basin and those with the boundary solution is not as clear as in S48's model. However, under the assumption of small damping, $\epsilon \ll 1$, the third term in the $\nabla^4$ operator [given by (7)] cannot be neglected for $\delta^4 \sim O(\epsilon^3)$. Similar to S48's model, the vorticity balance in the interior, which is determined in this limit by the interplay of 3 terms: $\dfrac{\partial \psi}{\partial x}$, $\dfrac{\partial^4 \psi}{\partial y^4}$ and $-\sin(\pi y)$, yields a $\delta-$dependent equatorward transport. This $\delta-$dependent transport in the interior is balanced by an equal, poleward-directed, transport along the western boundary. Although the vorticity associated with $\dfrac{\partial^4 \psi}{\partial y^4}$ is not as intuitive as that associated with $\dfrac{\partial^2 \psi}{\partial y^2}$ in S48's model the change it entails in Sverdrup's interior solution is similar.

The results derived here highlight an important effect that was overlooked in the classical/traditional WBC theory, namely, the effect of the domain aspect ratio on the Sverdrup solution of the inner basin which results from the meridional shear of the zonal velocity in a narrow zonal channel.

The non-dimensional formulation presented here does not alter the physical basis of the S48 and M50 models. We emphasize that the dimensional transport (calculated from the product of the non-dimensional transport and $\gamma \beta L_y^3 H_0$) in S48's model varies linearly with the Rayleigh friction coefficient ($r$) while in M50's model it is nearly independent of the eddy viscosity ($\mu$). In both models the transport is linear with magnitude of the wind-stress curl.

The application of our results to present-day ocean attributes the small transport of the EAC compared to the other WBCs to the geometry of the South Pacific ocean. In reality, factors other than the domain aspect ratio may also be important in determining the transport. For instance, the Brazil current's volumetric transport is low (especially in the northern part) because the current is largely confined to the continental shelf (Stramma et al., 1990). Temperature-driven buoyancy fluxes can also affect the transport of a WBC (Hogg and Gayen, 2020).

It is highly plausible that with a different arrangement of the continents in previous geologic times, the small domain aspect ratio that persisted in the ocean at that time could not support a strong WBC. Thus, the resulting higher pole to equator temperature gradient might have strongly affected the Meridional Overturning Circulation. This hypothesis should be addressed in a future work.

*Acknowledgements.* The authors thank D. Marshall and another anonymous reviewer whose comments were helpful in distilling some of the subtle points addressed in this work.

*Financial support.* This research was supported by the ISF-NSFC joint research program (grant number 2547/17).

*Code availability.* The numerical model used in this work can be downloaded from https://github.com/kaushalgianchandani/SWEsolver

*Data availability.* No data were used or generated in this theoretical research.

*Author contributions.* All authors contributed equally to this work.

*Competing interests.* The authors declare no conflict of interest.

## Appendix A: Typos in Stommel (1948)

There are some typos in the expression for $u$ [equation (21)] and $\eta$ [equation (23)] in Stommel (1948). The correct expressions
285  are given as:

$$u = \gamma(b/\pi)\cos(\pi y/b)\left(pe^{Ax} + qe^{Bx} - 1\right) \tag{A1}$$

$$\eta = -(F/gD)\cos(\pi y/b)(e^{Ax}p/A + e^{Bx}q/B)$$
$$-(f\gamma/g)(b/\pi)^2\sin(\pi y/b)(pe^{Ax} + qBe^{Bx} - 1)$$
$$+(\partial f/\partial y)(\gamma/g)(b/\pi)^3\cos(\pi y/b). \tag{A2}$$

290  For the reader's perusal, the variables in the aforementioned equations are the same as the ones defined in Stommel (1948).
Fig. A1 provides excerpts from Stommel (1948) over which, the corrections have been highlighted.

**Figure A1.** Corrections to $u$ and $h$ indicated over excerpts from Stommel (1948).

## Appendix B: Limiting cases of stream function $\psi$ in S48's model

In the limit $\epsilon \leq \delta^2$, the solution $\psi$ tends to:

$$\lim_{\epsilon \leq \delta^2} \psi(x,y) = c_1 \frac{\delta^2}{\epsilon \pi^2} \sin(\pi y)(x) \tag{B1}$$

where $c_1 = \lim_{\epsilon \leq \delta^2} A$ is a number $\ll 1$. On the other hand, in the limit of $\epsilon > \delta^2$, the solution $\psi$ becomes:

$$\lim_{\epsilon > \delta^2} \psi(x,y) = \frac{\delta^2}{\epsilon \pi^2} \sin(\pi y)[p(e^{c_2 x} + e^{c_2(1-x)}) - 1] \tag{B2}$$

where $c_2 = \lim_{\epsilon > \delta^2} A = \frac{\pi}{\delta}$ and $p = \frac{e^{c_2} - 1}{e^{2c_2} - 1}$. The function $\lim_{\epsilon > \delta^2} \psi(x,y)$ is symmetric about $x = \frac{1}{2}$.

## Appendix C: Zonal and meridional extents of the five western boundary currents in present-day world ocean

To determine the zonal and meridional extents of a basin containing a WBC, we identified the mean initiation and termination latitudes of each WBC based on the available literature. The Gulf Stream begins at the tip of Florida ($\sim 25°$ N) and runs upto $\sim 38°$ N where it breaks of into hot and cold rings (Hogg and Johns, 1995). The Kuroshio originates from the bifurcation of North Equatorial current at 12 - 13° N, although this bifurcation point can vary between 10 - 15° N (Qiu and Lukas, 1996); it separates from the Japan coast at 35° N as a meandering current colloquially known as the Kuroshio extension which stretches as far as $\sim 38°$ N (Kida et al., 2016). The East Madagascar-Agulhas current, in the South Indian ocean, runs from 20° S to 40° S (Lutjeharms et al., 1981; Gordon, 1985; Lutjeharms, 2006) — however the current retroflects between 38° S to 40° S (Quartly and Srokosz, 1993). Moreover, the African continental landmass ends close to 35° S. The Brazil current begins between 10° S and 12° S (Peterson and Stramma, 1991; Stramma et al., 1990) but the intense current attains its intense speed characteristic of a WBC only when it crosses the Vitoria-Trindade Ridge at 20.5° S (Evans et al., 1983). This current separates from the coastline at a mean value of 36° S $\pm$ 1.1° (Olson et al., 1988). The last of the five WBCs in the world ocean is the East Australian Current (EAC) that extends from 18° S to around 35° S (Boland and Church, 1981; Ridgway and Godfrey, 1994) but a characteristic southward flow is evident only when EAC crosses 22° S (Ridgway and Dunn, 2003); the current usually separates from the coast at 33° S (Archer et al., 2017).

We define the meridional extent ($L_y$) as the distance between the initiation and termination latitudes of the WBC. On the other hand, to determine the zonal extent ($L_x$) we calculate the distances between the land masses at both the initiation latitude and termination latitude. The average of the two distances is defined as the typical $L_x$ for a given WBC. For instance, the approximate initiation and termination coordinates for the Kuroshio are 13° N, 125° E and 35° N, 140 E respectively, which yields $L_y \approx 2500$ km. The distances to the opposite landmass, the North American continent (which forms the eastern boundary of the basin) as calculated from the initiation coordinate and termination coordinate are $\sim 9000$ km and $\sim 15000$ km respectively. Thus, the typical zonal extent of the basin is assumed to be 12000 km.

The mean dimensions $L_x$ and $L_y$ for all the five WBCs in the world ocean are given by Table 1. The 'error' in $L_y$ accounts for the variation between different references of the initiation and termination latitudes and the error in $L_x$ is the deviation

**Table C1.** Dimensions of the gyres that contain the five western boundary currents in the present-day world ocean.

| Current | Western edge of the basin | | Eastern edge of the basin | | Basin's dimensions | |
| --- | --- | --- | --- | --- | --- | --- |
| | Initiation | Termination | Initiation | Termination | Zonal ($L_x$) | Meridional ($L_y$) |
| Gulf Stream | 25° N 80° W | 38° N 75° W | 25° N 16° W | 38° N 10° W | 6000 ± 400 km | 1500 ± 200 km |
| Kuroshio | 13° N 125° E | 35° N 140° E | 13° N 92° W | 35° N 121° W | 12000 ± 3000 km | 2500 ± 400 km |
| Madagascar-Agulhas | 20° S 50° E | 35° S 20° E | 20° S 116° E | 35° S 116° E | 7500 ± 800 km | 1700 ± 350 km |
| Brazil | 21° S 40° W | 35° S 54° W | 21° S 13° E | 35° S 19° E | 6000 ± 400 km | 1600 ± 500 km |
| East Australian | 22° S 150° E | 33° S 152° E | 22° S 70° W | 33° S 72° W | 12500 ± 2000 km | 1200 ± 250 km |

of the measured zonal distances along initiation and termination latitude from the mean value. Based on these values of $L_x$ and $L_y$ a range of parameters damping ($\alpha$) and domain aspect ratio ($\delta$) corresponding to every WBC was estimated and these values of $\alpha$ and $\delta$ were employed it to distinguish between the five WBCs. Typical values of $L_x$ and $L_y$ for the ocean basins that contain the WBCs were also estimated using the mean streamlines in the ocean as calculated by Maximenko et al. (2009) — these values were well within the range cited in Table C1.

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
