# Peer review of "On the role of domain aspect ratio in the westward intensification of wind-driven surface ocean circulation"

_Ocean Science, 2020_

## Referee Comment (RC1) · Anonymous Referee #1 · 27 Oct 2020

The authors revisit the classical linear theories of the western boundary current by Stommel (1948) and Munk (1950) to examine the role of the domain aspect ratio. A non-dimensional linear vorticity equation is derived to show the sensitivity of the western boundary current transport to domain aspect ratio (delta) and drag coefficient (alpha) and then apply the results to explain the weak East Australian Current of the South Pacific. While I agree that non-dimensional equations are useful, it is unclear what the new physical findings of this study are. Is there a change in how the vorticity balances in the western boundary layer? The authors need to clarify that the parameter dependence of the WBC solution is not just a result of the mathematical formulation that the authors have chosen.

[Figure]

(1) I would like the authors to discuss the sensitivity of the results to the choice of the boundary current width. In terms of mass balance, the WBC simply returns the Sverdrup interior so if the Sverdrup interior is kept constant, the transport of the WBC will not change. The meridional velocity at the western boundary also varies differently in the zonal direction for S48 and M50: for S48, it decays exponentially with epsilon while for M50, a maximum occurs near epsilon. The way the transports are estimated (Equations 4 and 8) does not seem to fully take these differences into account.

(2) The scaling of the stream function depends on delta [gamma*beta*Ly^3 = tau*pi/(rho*Ho*beta)*delta^-1]. Is the sensitivity of the WBC transport to delta a consequence of using such a scaling? As Ly changes, so do the magnitude of the wind stress curl and the scaling of the stream function. What is the benefit of using such scaling? To focus on the WBC, isn't it better to keep the wind stress curl constant and keep the Sverdrup interior the same?

(3) Figure 4 shows that the transport of the East Australian Current (EAC) is weaker than the other WBCs because of the small delta. But how was Ly determined for EAC? The meridional scale of this western boundary current appears to be different from the spatial scale of the winds. Zero wind stress curl does not exist around 22S (e.g. https://booksite.elsevier.com/DPO/chapterS10.html).

———————————————————

---

## Referee Comment (RC2) · David P. Marshall (Referee) · 12 Nov 2020

This manuscript revisits the classical wind-driven gyre models of Stommel (1948) and Munk (1950). After non-dimensionalising the underlying equations and writing down the analytic solutions, the authors derive estimates of the frictional boundary current transports. They conclude that the transport scales with the linear drag coefficient in the Stommel model, but not with the lateral viscosity ("damping factor") in the Munk model. They also comment on the scaling of the boundary current transports with the domain aspect ratio.

While it was an enjoyable exercise to revisit the general solutions of these classical

models, I'm afraid that I am unable to recommend the manuscript for publication as I believe the results are misleading, at least in the parts of parameter space of most relevance to the ocean (and it is debatable that these models are of any quantitative relevance beyond their substantial conceptual value).

*Major comment:*

The solution (3) to the Stommel model is indeed the most general, but this form rather obscures the essential physics in the physically-relevant limit of small $\alpha$ (i.e., the boundary current width is much smaller than the basin width). The authors erroneously state that in this limit, the solution becomes linear in $x$ and can satisfy just one boundary condition. However, a more careful expansion of the exponential terms leads to a more complete solution.

I prefer to see this by assuming $\alpha$ is small and hence $\partial^2/\partial x^2 \gg \partial^2/\partial y^2$. Thus (1) is well approximated by

$$\alpha \frac{\partial^2 \psi}{\partial x^2} + \frac{\partial \psi}{\partial x} \approx \sin(\pi y),$$

the solution to which is

$$\psi \approx \left(x - 1 + e^{-\alpha x}\right) \sin(\pi y).$$

This solution consists of the Sverdrup (1947) solution in the basin interior – the first two terms in brackets on the right-hand side – and a Stommel (1948) western boundary current correction – the third tern in brackets on the right-hand side.

Mathematically, the dropping of the $\alpha \partial^2 \psi / \partial y^2$ term in (1) means that the particular integral is instead formed by balancing $\partial \psi / \partial x$ against the wind stress curl on the right-hand side. However, the same result can be obtained through a careful treatment of the limit of small $\alpha$ in the two exponential terms in the more general solution.

The implications are that the western boundary current transport is:

- approximately equal (and opposite) to the Sverdrup gyre transport;

- independent of the linear drag coefficient, $\alpha$;

- independent of the basin aspect ratio, $\delta$.

These conclusions are at odds with those stated in the manuscript.

I do accept that in the case that $\alpha$ becomes larger, the boundary current transport, and indeed the entire nature of the solution changes, but it is hard to see what relevance this has to a large-scale ocean basin.

*Minor comments:*

1. I don't understand why the authors estimate the boundary current transport rather than simply calculate the maximum value of the streamfunction which gives the actual western boundary transports. If I follow correctly, the authors also invoke the Stommel scaling for the boundary layer width, but that only holds in the low $\alpha$ limit.

2. It might be helpful to many readers to state the original equations, before non-dimensionalising.

3. $\gamma$ is the non-dimensional magnitude of the wind stress curl, not the wind stress.

4. In figure 1(d), why is the eastern boundary condition not satisfied?

5. I'm really struggling with the numerical and theoretical boundary current transport scalings in figure 3, especially the upper panel for the Stommel gyre. I understand that the authors will state that these results support their conclusions, but they are at odds with the basic dynamics of the low $\alpha$ limit (see major comment above). The explanation in lines 194-189 of what has been done to obtain the theoretical scalings, and why, is confusing (to me at least).

6. Regarding figure 4, are you seriously suggesting that $\alpha = 0.5$ is an appropriate value for the East Australian Current? This would imply the failure of geostrophy, for example.

[Figure]

7. Following on from point 6, there are numerous other processes that are likely in influence the width of real world western boundary currents ahead of linear bottom drag and lateral friction. These include relative vorticity (Fofonoff, 1954; Charney, 1955), stratification (the deformation radius emerges as a natural length scale), bottom topography (e.g., Hughes and de Cuevas, 2001), eddy fluxes (e.g., Eden and Olbers, 2010).

*References*

Charney, J.G., 1955. The Gulf Stream as an inertial boundary layer. Proc. Natl. Acad. Sci. U.S.A. 41, 731.

Eden, C., Olbers, D., 2010. Why western boundary currents are diffusive: a link between bottom pressure torque and bolus velocity. Ocean Model. 32, 14–24.

Fofonoff, N.P., 1954. Steady flow in a frictionless homogeneous ocean. J. Mar. Res. 13, 254–262.

Hughes, C.W., de Cuevas, B.A., 2001. Why western boundary currents in realistic oceans are inviscid: a link between form stress and bottom pressure torques. J. Phys. Oceanogr. 31, 2871–2885.

Sverdrup, H.U., 1947. Wind-driven currents in a baroclinic ocean with application to the equatorial current in the eastern pacific. Proc. Natl. Acad. Sci. U.S.A. 33, 318–326.

---

## Referee Comment (RC3) · David P. Marshall (Referee) · 13 Nov 2020

Sorry, in point 5, this should have read "lines 184-189".

———————————————————

---

## Editor Comment (EC1) · Katsuro Katsumata (Editor) · 30 Nov 2020

Thank you for your replies to the reviewers' comments.

I find the authors' rebuttal to the major point raised by Reviewer 2 is valid and look forward to a revised manuscript reflecting other comments from both reviewers.

Re: minor comment 2 from Reviewer 2, I am inclined to the opinion of the reviewer that a paragraph showing the nondenominational form would save inexperienced or interdisciplinary readers looking up yet another reference. At least the authors could explicitly name a couple of references in the text.

[Figure]

Please note that I have been notified that the very first sentence of the manuscript closely resembles that of S48. The authors' intention might be a homage to the classic paper, but to avoid unnecessary misunderstandings I would suggest the authors rewrite this sentence.

———————————————————

---

## Author Response (AR1)

In response to the comments we received from the editor and from the reviewers we have re-written most of the manuscript including:

1. Adding the dimensional vorticity equation

2. Re-writing the entire text with  $\epsilon$  as the non-dimensional friction coefficient instead of  $\alpha$

3. Producing new versions of the four figures including new simulations with fixed values of the new friction parameter

4. Adding a detailed description in the Discussion regarding the ramifications of our results to the Sverdrup solution in the interior of the basin (that drives the poleward directed transport in the WBC)

A point-by-point response to the particular comments, including pointers to the text where these changes were implemented follows. A marked up manuscript version in which the differences between the new and previous versions are clearly marked is uploaded as a separated file.

Subject: Authors' response to RC1 and corresponding changes to the MS.

We thank the referee for helping us improve the quality of our paper. In the following we address the minor comments raised in the review.

While I agree that non-dimensional equations are useful, it is unclear what the new physical findings of this study are. Is there a change in how the vorticity balances in the western boundary layer? The authors need to clarify that the parameter dependence of the WBC solution is not just a result of the mathematical formulation that the authors have chosen.

Response: While the role of damping ( $\alpha$  or  $\epsilon$  in our formulation) in the westward intensification has been previously discussed extensively in the literature, the dependence of the WBC's transport on the domain aspect ratio has not been studied (save for its mention in Bye and Veronis, 1979). The novel finding of our study is the first quantification of the dependence of the WBC's transport on the domain aspect ratio. This finding enables, in turn, its application to the five known WBCs.

Clearly, the vorticity balance in the boundary layer is unaffected by our scaling but in the interior of the basin our formulation and scaling shows that the term of the Laplacian proportional to  $\partial^2 \psi / \partial y^2$  can be neglected only for  $\delta \geq 1$  (see our detailed response to the next comment). Our concise formulation underscores features that exist in the dimensional formulation (as in our numerical simulations) but are hard to see when dealing with five model parameters.

(1) I would like the authors to discuss the sensitivity of the results to the choice of the boundary current width. In terms of mass balance, the WBC simply returns the Sverdrup interior so if the Sverdrup interior is kept constant, the transport of the WBC will not change. The meridional velocity at the western boundary also varies differently in the zonal direction for S48 and M50: for S48, it decays exponentially with epsilon while for M50, a maximum occurs near epsilon. The way the transports are estimated (Equations 4 and 8) does not seem to fully take these differences into account.

Response: It is indeed correct that the WBC's transport is equal in magnitude to the Sverdrup transport in the basin's interior. However, the Sverdrup transport itself is dependent on the basin's aspect ratio. The correction to the Sverdrup transport in bounded domains was developed in Bye and Veronis (1979).

In terms of our scaling this correction can be derived by noting that since  $\alpha = \epsilon/\delta^2$ , equation (1) of the manuscript implies:

$$\left(\epsilon \frac{\partial^2}{\partial x^2} + \frac{\epsilon}{\delta^2} \frac{\partial^2}{\partial y^2}\right)\psi + \frac{\partial\psi}{\partial x} = \sin(\pi y)$$

where  $\epsilon = (r/\beta L_x)$  is a proxy of damping and the non-dimensional width of the WBC. Under the assumption of small damping i.e.  $\epsilon \ll 1$ , the term  $\frac{\epsilon}{\delta^2} \frac{\partial^2 \psi}{\partial y^2}$ , cannot be neglected in the interior solution when  $\delta^2 \sim O(\epsilon)$  i.e. the Sverdrup balance becomes a function of  $\delta$  in this case. As was rightly pointed out by the referee, the WBC 'simply returns' this  $\delta$ -dependent Sverdrup transport.

We employed the simple scaling to obtain our ad-hoc definition of the boundary layer width [i.e.  $\epsilon = r/(\beta L_x)$  for Stommel's model and  $\epsilon = [\mu/(\beta L_x^3)]^{(1/3)}$  for Munk's model] to make the paper more accessible to oceanographers that are inclined towards observations or numerical

modeling. Equations (4) and (8) are of the form:

$$Tr = \frac{\delta}{\alpha \pi^2} [1 - O(\epsilon)] \qquad (4')$$

and

$$Tr = \delta[1 - O(\epsilon)] \qquad (8')$$

These expressions are valid for values of  $\epsilon$  for which a WBC exists and they show that our results are not sensitive to the precise definition of the WBC's width. For instance, the width of the WBC can also be defined as the value of x for which the stream function reaches an extrema. By this definition,  $\epsilon_S \sim 5\epsilon$  for Stommel's and  $\epsilon_M \sim 2\epsilon$  for Munk's model, where  $\epsilon$  is the current definition of the WBC's width in the two models. The respective transports in the two cases are given by:

$$Tr_S = \frac{\delta}{\alpha \pi^2} (1 - p e^{5A\epsilon} - q e^{5B\epsilon})$$

and

$$Tr_M = \delta \left( 1 - e^{-1} \left[ \cos(\sqrt{3}) + \frac{1 - 2\epsilon}{\sqrt{3}} \sin(\sqrt{3}) \right] \right)$$

Here, we see that the two transports calculated by the new definition of WBC's width are also of the form (4') and (8'), which indicates that the results presented in this paper are independent of the precise definition of WBC's width. We thank the referee for this comment and we will further highlight the independence of our results to the choice of WBC's width in the revised manuscript.

Changes: Lines 121 - 126 in the revised version of the manuscript discuss how our results are independent of the definition of the WBC's width.

(2) The scaling of the stream function depends on delta  $[\gamma\beta L_y^3 = \tau\pi/(\rho H_0\beta\delta)]$ . Is the sensitivity of the WBC transport to  $\delta$  a consequence of using such a scaling? As  $L_y$  changes, so do the magnitude of the wind stress curl and the scaling of the stream function. What is the benefit of using such scaling? To focus on the WBC, isn't it better to keep the wind stress curl constant and keep the Sverdrup interior the same?

Response: No, the sensitivity of the WBC's transport to  $\delta$  is not a consequence of using our particular scaling. To appreciate this subtle dependence one should compare a square basin, where  $\delta = 1$ , with a "narrow and long channel-like" basin where  $\delta \ll 1$ . In a square basin, the classical approach of equating  $\partial \psi / \partial x$  to  $\sin(\pi y)$  works well since the North-South gradient of the zonal velocity (represented by  $\partial^2 \psi / \partial y^2$ ) is small and can be neglected from the interior solution. However, in a "channel-like" ocean this quantity is large and cannot be neglected from the balance of terms in the interior solution. Surely, an examination of the 3 vorticity terms in the interior  $(\partial \psi / \partial x, \text{ wind-stress and } \partial^2 \psi / \partial y^2)$  clarifies that the WBC in the "channel-like" ocean should be weaker compared to a square ocean.

As in all non-dimensional problems, the choice of scaling is not unique. We choose this scaling to stay consistent with the one proposed in Bye and Verionis, 1979. The results are independent of magnitude of wind-stress curl because the differential operators in the vorticity equations of Stommel and Munk are all linear.

We thank the reviewer for this comment and we include the aforementioned example in the revised manuscript to further elaborate the conceptual aspect of our paper. We will also re-write the paper with  $\epsilon$  as the damping parameter (instead of  $\alpha$ ) in both Stommel and Munk models. Changes: The entire paper was re-written with  $\epsilon$  as the damping parameter (instead of  $\alpha$ ). Furthermore, lines 232 - 262 in the revised version of the manuscript highlight why the conceptual aspect of this paper is not a consequence of the scaling employed. This part of the discussion underscores the 'physical' meaning of the vorticity terms in the interior of the basin in both models and presents our results in a more intuitive framework.

(3) Figure 4 shows that the transport of the East Australian Current (EAC) is weaker than the other WBCs because of the small delta. But how was Ly determined for EAC? The meridional scale of this western boundary current appears to be different from the spatial scale of the winds. Zero wind stress curl does not exist around 22S (e.g. https://booksite.elsevier.com/DPO/chapterS10.html)

Response: The conflict between the geometry of the ocean basin and the overlying wind stress in the WBCs is independent of the model used for explaining the properties of the WBCs and hence does not affect our formulation and scaling. In appendix C of our paper we discuss in detail how the irregular shaped basin in the world ocean were approximated to obtain values of  $L_x$  and  $L_y$ . The error-bars along the ordinate provide a range between which  $\delta$  can vary for different choices of  $L_y$  and  $L_x$ .

Changes: No changes were made per this minor comment because addressing the issue highlighted by the referee in this comment is beyond the scope of this study.

**References:**

Bye, J. A. T., and George Veronis. "A correction to the Sverdrup transport." Journal of Physical Oceanography 9.3 (1979): 649-651.

**Subject: Authors' response to RC2 & RC3 and the corresponding changes to the MS.**

We thank the referee for helping us improve the quality of our paper. However, the referee's (single) major comment is completely irrelevant to our analysis of Stommel's model. Furthermore, the recommendation to reject our manuscript completely ignores our (unchallenged) analysis of Munk's model.

While it was an enjoyable exercise to revisit the general solutions of these classical models, I'm afraid that I am unable to recommend the manuscript for publication as I believe the results are misleading, at least in the parts of parameter space of most relevance to the ocean (and it is debatable that these models are of any quantitative relevance beyond their substantial conceptual value).

We believe that the values we selected for the non- dimensional parameters in Stommel's and Munk's model fall within the relevant ranges of values of the dimensional parameters (see our detailed response to minor comment #6). We agree that our paper has two foci - one "conceptual" and the other "quantitative" (i.e. related to the world ocean).

**Major comment:**

The solution (3) to the Stommel model is indeed the most general, but this form rather obscures the essential physics in the physically-relevant limit of small  $\alpha$  (i.e. the boundary current width is much smaller than the basin width). The authors erroneously state that in this limit, the solution becomes linear in x and can satisfy just one boundary condition. However, a more careful expansion of the exponential terms leads to a more complete solution.

I prefer to see this by assuming  $\alpha$  is small and hence  $\frac{\partial^2}{\partial x^2} \gg \frac{\partial^2}{\partial y^2}$ . Thus (1) is well approximated by:

$$\alpha \frac{\partial^2 \psi}{\partial x^2} + \frac{\partial \psi}{\partial x} \approx \sin(\pi y)$$

the solution to which is

$$\psi \approx (x - 1 + e^{-\alpha x})\sin(\pi y)$$

This solution consists of the Sverdrup (1947) solution in the basin interior – the first two terms in brackets on the right-hand side – and a Stommel (1948) western boundary current correction – the third term in brackets on the right-hand side.

Mathematically, the dropping of the  $\alpha \frac{\partial^2}{\partial x^2}$  term in (1) means that the particular integral is instead formed by balancing  $\partial \psi / \partial x$  against the wind stress curl on the righthand side. However, the same result can be obtained through a careful treatment of the limit of small  $\alpha$  in the two exponential terms in the more general solution.

The implications are that the western boundary current transport is:

- approximately equal (and opposite) to the Sverdrup gyre transport
- independent of the linear drag coefficient,  $\alpha$ ;
- independent of the basin aspect ratio,  $\delta$ .

These conclusions are at odds with those stated in the manuscript. I do accept that in the case that  $\alpha$  becomes larger, the boundary current transport, and indeed the entire nature of the solution changes, but it is hard to see what relevance this

**has to a large-scale ocean basin.**

Response: The assumption,  $\frac{\partial^2}{\partial x^2} \gg \frac{\partial^2}{\partial y^2}$  that underlies the referee's comment, implies that the general solution of the associated homogeneous equation is y-independent i.e. the y-dependence of the solution is identical to that of the inhomogeneous forcing term. In our model, this assumption translates to  $\frac{1}{\delta^2} = 0$  in the Laplacian  $\nabla^2 = \delta^2 \frac{\partial^2}{\partial x^2} + \frac{\partial^2}{\partial y^2}$  [given by (2) in the manuscript]. While this y-independent limit can yield a fast-flowing WBC in Stommel's model, it completely undermines the role of basin's aspect ratio ( $\delta$ ) in determining the transport of the WBC, which is the main sermon of our paper (as is evident from the paper's title). For large scale circulation, typical values of  $\delta < 0.5$  yield  $\frac{1}{\delta^2} > 1$  i.e. setting  $\frac{1}{\delta^2} = 0$  is inconsistent with the intended "quantitative" applications. The alternative is to employ the general solution of equation (1) without assuming y-independence of the Laplacian, which is precisely what we did in our paper.

Moreover, Bye and Veronis (1979) succinctly established the  $\delta$ -dependence of the Sverdrup transport. In terms of our scaling this correction can be derived by noting that since  $\alpha = \epsilon/\delta^2$ , equation (1) of the manuscript implies:

$$\left(\epsilon \frac{\partial^2}{\partial x^2} + \frac{\epsilon}{\delta^2} \frac{\partial^2}{\partial y^2}\right)\psi + \frac{\partial\psi}{\partial x} = \sin(\pi y)$$

where  $\epsilon = (r/\beta L_x)$  is a proxy of damping and the non-dimensional width of the WBC. Under the assumption of small damping i.e.  $\epsilon \ll 1$ , the term  $\frac{\epsilon}{\delta^2} \frac{\partial^2 \psi}{\partial y^2}$ , cannot be neglected in the interior solution when  $\delta^2 \sim O(\epsilon)$  i.e. the Sverdrup balance becomes a function of  $\delta$  in this case. The latter two bullet points that the referee makes lead to the unacceptable result that the strength of the WBC (that is equal in magnitude to the  $\delta$ -dependent Sverdrup transport) is not determined by either of the model parameters  $\alpha$  (or  $\epsilon$ ) and  $\delta$ !

To appreciate this subtle issue one should compare a square basin, where  $\delta = 1$ , with a "narrow and long channel-like" basin where  $\delta \ll 1$ . In a square basin, the classical approach of equating  $\partial \psi / \partial x$  to  $\sin(\pi y)$  works well since the North-South gradient of the zonal velocity (represented by  $\partial^2 \psi / \partial y^2$ ) is small and can be neglected from the interior solution. However, in a "channel-like" ocean this quantity is large and cannot be neglected from the balance of terms in the interior solution. Surely, an examination of the 3 vorticity terms in the interior  $(\partial \psi / \partial x, wind-stress and \partial^2 \psi / \partial y^2)$  clarifies that the WBC in the "channel-like" ocean should be weaker compared to a square ocean. Clearly, in the referee's approach there is no difference between the two oceans.

Here, we take the opportunity to thank the referee for his comment. To reconcile our approach with the existing literature, we will re-write the paper with  $\epsilon$  as the parameter for damping (instead of  $\alpha$ ). We will also include the aforementioned comparison between a square and "channel-like" basin to further emphasize the conceptual aspect of our study.

Changes: The entire paper was re-written with  $\epsilon$  as the parameter for damping (instead of  $\alpha$ ). Lines 232 - 262 in the revised version of the manuscript underscore the 'physical' meaning of the vorticity terms in the interior of the basin in both models. This further adds to the conceptual aspect of this paper and presents our results in a more intuitive framework.

**Minor comments:**

1. I don't understand why the authors estimate the boundary current transport rather than simply calculate the maximum value of the streamfunction which gives the actual western boundary transports. If I follow correctly, the authors also invoke the Stommel scaling for the boundary layer width, but that only holds

**in the low $\alpha$ limit**

Response: The reviewer is right in that a simpler definition could have been used to estimate the transport. However, the definition based on the maximum point of the stream-function gives a width of about 500 km in 10,000 km basin while our definition gives a width of about 100 km in the same basin. Moreover, using Stommel's scaling to obtain our ad-hoc definition of the boundary layer width [i.e.  $\epsilon = r/(\beta L_x)$ ] makes the paper more accessible to oceanographers that are inclined towards observations or numerical modeling.

The expression for transport [given by (4) in the manuscript] is valid for the referee's definition of boundary layer width as well and the choice of the WBC's width does not alter the conclusions presented in our paper. We thank the referee for bringing this to our attention. In the revised manuscript, we will further emphasize that the results are not sensitive to the choice of WBC's width.

Changes: Lines 121 - 126 in the revised version of the manuscript discuss how our results are independent of the definition of the WBC's width.

**2. It might be helpful to many readers to state the original equations, before nondimensionalising.**

Response: We will accept an editorial decision on this matter but since both forms of the vorticity equation – dimensional and non-dimensional – appear in so many textbooks and research papers we thought that presenting both versions is redundant.

Changes: We accepted the editorial decision and included the original dimensional equation - see equation (1) (line 73) in the revised version of the manuscript.

**3. $\gamma$ is the non-dimensional magnitude of the wind stress curl, not the wind stress**

Response: We thank the referee for pointing this out. We will correct this in the revised manuscript.

Changes: Appropriate corrections were made in lines 79 and 248 of the revised manuscript.

4. In figure 1(d), why is the eastern boundary condition not satisfied? Response: Figure 1(d), depicts the analytically obtained, non-dimensional streamfunction for Munk's model. In Munk's model the stream function, given by (7), is a function of  $|\alpha| = \mu \frac{L_x}{\beta L_y^4}$ and does not vanish identically even for small  $|\alpha|$  at either boundary (although the values of the streamfunction at the boundaries are rather small). For large values of  $|\alpha|$ , the value of the streamfunction at the boundary is no longer close to 0 (as it is for smaller  $|\alpha|$ ) and we see the less than optimal behaviour as depicted in Figure 1(d).

We thank the referee for his input and will emphasize this further in the revised manuscript.

Changes: Lines 145-147 discuss the subtlety of why the analytic stream function for M50's model is not well behaved for large  $\epsilon$ . We further underscore the crudeness of the analytic stream function in line 203.

5. I'm really struggling with the numerical and theoretical boundary current transport scalings in figure 3, especially the upper panel for the Stommel gyre. I understand that the authors will state that these results support their conclusions, but they are at odds with the basic dynamics of the low  $\alpha$  limit (see major comment above). The explanation in lines 194-189 of what has been done to obtain the theoretical scalings, and why, is confusing (to me at least).

Response: The results shown in Figure 3 highlight the consistency of our theoretical/analytic findings with (dimensional!) numerical simulations. Our response to the major comment above provides the explanation of the consistency between our dimensional numerical simulations and the non-dimensional analysis based on our scaling.

Changes: Figure 3 was re-drawn with  $\epsilon$  as the abscissa and the results are consistent with our findings.

**6. Regarding figure 4, are you seriously suggesting that $\alpha = 0.5$ is an appropriate value for the East Australian Current? This would imply the failure of geostrophy, for example.**

Response: The parameter  $\alpha = rL_x/(\beta L_y^2)$  for each WBC was estimated by substituting  $\beta = 2 \times 10^{-11} \text{ m}^{-1} \text{s}^{-1}$ , Rayleigh friction coefficient  $r = 1/10 \text{ (days)}^{-1}$  and the typical dimensions  $(L_x \text{ and } L_y)$  of the basin. Other choices of the Rayleigh friction coefficient do not alter the results. Figure 1 below depicts the results for  $r = 1/20 \text{ (days)}^{-1}$  and the same values of  $\beta$ ,  $L_x$  and  $L_y$ .

We thank the referee for suggesting this. If requested, we will be happy to include the attached figure in the revised manuscript.

Figure 1: Alternative figure to panel (a) of Figure 4

Changes: The alternative figure above was not used because the manuscript was rewritten with  $\epsilon$  as the damping parameter. Figure 4 was re-drawn with  $\epsilon$  as the abscissa and the new figure is presented in the revised manuscript.

7. Following on from point 6, there are numerous other processes that are likely in influence the width of real world western boundary currents ahead of linear bottom drag and lateral friction. These include relative vorticity (Fofonoff, 1954; Charney, 1955), stratification (the deformation radius emerges as a natural length scale), bottom topography (e.g., Hughes and de Cuevas, 2001), eddy fluxes (e.g., Eden and Olbers, 2010).

Response: We agree. However, none of these works addressed the role of basin's aspect ratio in determining the transport of the WBC.

Changes: No changes were made per this minor comment because addressing the issue highlighted by the referee in this comment is beyond the scope of this study.

Correspondence: Nathan Paldor (nathan.paldor@mail.huji.ac.il)

**Abstract.** The two seminal studies on westward intensification, carried out by Stommel and Munk over 70 years ago, are revisited to elucidate the role of the domain aspect ratio (i.e. meridional to zonal extents of the basin) in determining the transport of the western boundary current (WBC). We examine the general mathematical properties of the two models by transforming them to differential problems that contain only two parameters — the domain aspect ratio and the non-dimensional

- 5 damping (viscous) coefficient. Explicit analytical expressions are obtained from solutions of the non-dimensional vorticity equations and verified by long-time numerical simulations of the corresponding time-dependent equations. The analytical expressions as well as the simulations, imply that in Stommel's model both the domain aspect ratio and the damping parameter contribute equally to the non-dimensional transport of the WBC. However, the transport increases as a cubic power in the aspect ratio and decreases linearly with the damping coefficient. On the other hand, in Munk's model the WBC's transport
- 10 varies linear increases linearly with the domain aspect ratio, while the damping parameter coefficient plays a minor role only. This finding is employed to explain the weak WBC in the South Pacific. The decrease in transport of the WBC for small domain aspect ratio results from the decrease in Sverdrup transport in the basin's interior because the meridional shear of the zonal velocity cannot be neglected as an additional vorticity term.

Copyright statement. TEXT

**15 1 Introduction**

As was noted by Henry Stommel, in the opening sentence of his seminal 1948 study "Perhaps the most striking <del>characteristic of</del> the surface circulation in an ocean basin is the east-west asymmetry: feature of the general oceanic wind-driven circulation is the intense crowding of streamlines near the western borders of the oceans." These strong and narrow <del>pole-ward directed</del> <del>currents</del> of the oceans." These strong and narrow <del>pole-ward directed</del> <del>currents</del> of the oceans." (WBCs)<del>flow along the western</del>

[revised manuscript text omitted]
{1}{2\epsilon} + \frac{\pi}{\delta}\frac{\sqrt{1 + \frac{1}{4\pi^2\alpha^2\delta^2}}}{\sqrt{1 + \frac{\lambda^2}{4\pi^2\epsilon^2}}},$$
$$B = -\frac{1}{2\alpha\delta^2}\frac{1}{2\epsilon} - \frac{\pi}{\delta}\frac{\sqrt{1 + \frac{1}{4\pi^2\alpha^2\delta^2}}}{\sqrt{1 + \frac{\lambda^2}{4\pi^2\epsilon^2}}}.$$

As is evident from (4), the spatial structure of the stream function is controlled by both  $\alpha \in$  and  $\delta$ . Fig. 1Panels (a) and (c) 115 of Fig. 1 depict the stream function for two  $\alpha$  functions for two  $\epsilon$ -regimes of S48's model: (i) weak damping [ $\alpha \le O(1) \le \delta^2$ ] and (ii) strong damping [ $\alpha \ge O(1) \le \delta^2$ ]. For  $\alpha \le O(1) \le \delta^2$ , the solution  $\psi$  given by (4) becomes linear in x and thus can satisfy only one boundary condition out of two. This solution is commonly assumed to approximate the exact solution for  $\psi$  in the frictionless interior of the basin while a different approximation applies in the narrow, frictional, boundary layer adjacent to x = 0. Fig. 1(a) depicts this narrow boundary layer for  $\alpha = 0.1 \epsilon = 0.1 \delta^2$  where the stream function first decreases fast with 120 x at small x and then increases slowly with x for large x. In the range of  $\alpha \ge O(1)$ For  $\epsilon \ge \delta^2$ , the solution,  $\psi$ , is symmetric about  $x = \frac{1}{2}$  and can satisfy the two boundary conditions,  $\psi(0, y) = 0 = \psi(1, y)$ . This is demonstrated in the symmetric stream function depicted in Fig. 1(c) for  $\alpha = 10\epsilon = 10\delta^2$ . The explicit expressions of  $\psi$  in the two ranges of  $\alpha \in$  are given in the Appendix B.

---

## Author Response (AR2)

In response to the last round of review by referee2 and the editorial decision letter we've extended the range of $\varepsilon$ in panel (a) of Figure 4 to include the range well below $1\cdot10^{-3}$ and the panel now cover the range of between $0.6\cdot10^{-3}$ and $5\cdot10^{-3}$.

Corresponding changes were also made in the text (line 207) and in the caption of Figure 4.